# Hydrogen Bonds as Stability-Controlling Elements of Spherical Aggregates of ZnO Nanoparticles: A Joint Experimental and Theoretical Approach

**DOI:** 10.3390/ma16134843

**Published:** 2023-07-05

**Authors:** Ankica Šarić, Ines Despotović

**Affiliations:** 1Division of Materials Physics, Ruđer Bošković Institute, Centre of Excellence for Advanced Materials and Sensing Devices, Bijenička 54, HR-10002 Zagreb, Croatia; 2Division of Physical Chemistry, Ruđer Bošković Institute, Bijenička 54, HR-10002 Zagreb, Croatia

**Keywords:** ZnO nanoparticles, scanning electron microscopy, density functional theory (DFT), hydrogen bonds, diethanolamine, ethanolamine

## Abstract

The effects of various organic additives, such as diethanolamine (DEA) and ethanolamine (EA), and variations in aging times on the formation and stability mechanisms of spherical aggregates of ZnO nanoparticles (NPs) prepared by using solvothermal synthesis were studied. The experimental results of the structural, morphological and optical properties monitored by using X-ray diffraction, field-emission scanning electron microscopy (FE-SEM) and UV-Vis spectroscopy were supported by quantum chemical calculations at the level of density functional theory (DFT). Understanding the mechanism of spherical ZnO aggregate formation and its stability by mimicking the processes at the computer level was achieved through theoretical simulations of the ZnO surface/additive interactions using (ZnO)_36_–DEA and (ZnO)_36_–EA models. The fine-tuned spherical aggregation of ZnO nanoparticles was driven by various interactions, in particular, strong O–H∙∙∙O and weak N–H∙∙∙O hydrogen bonds as controlling interactions. The calculated negative free release energy, ∆G*_INT_, indicates that the ZnO surface/additive interaction in diethanolamine media is a spontaneous exergonic process (∆G*_INT_ = −7.73 kcal mol^−1^), whereas, in ethanolamine media, it is an unfavorable, slightly endergonic process (∆G*_INT_ > 0). The presence of two strong O–H∙∙∙O hydrogen bonds and, at the same time, a weaker N–H∙∙∙O hydrogen bond is the key factor for the very good and long-term aggregate stability of ZnO NPs in DEA media. This integrated experimental–theoretical study highlights the stability and compactness of spherical ZnO aggregates of ZnO NPs, prepared in the presence of diethanolamine compared to ethanolamine media, and provides a promising method and flexible design of ZnO nanomaterials to improve their adsorptive and optical properties.

## 1. Introduction

A unique combination of optical, catalytic, semiconducting and piezoelectric properties [1,2,3] combined with high thermal and chemical stability as a prerequisite results in superior functionality of zinc oxide (ZnO) [4,5]. Indeed, high thermal and mechanical stability and photostability are necessary conditions for the use of ZnO nanoparticles (NPs) as photocatalysts [6,7], while particle microstructure and morphology play a key role in optimizing the properties of shape-selected ZnO particles, which are crucial for adsorptive and photocatalytic efficiency. Certainly, zinc oxide has attracted worldwide research interest due to its low cost and toxicity, biocompatibility and diversity of morphologies [8,9,10,11]. Due to its low cost, high activity and low toxicity, ZnO is becoming the gold standard in photocatalytic wastewater treatment [12]. At the same time, ZnO can be used as a biocompatible, antiviral and antibacterial agent, which enables its use in biomedical applications [13,14]. Encouraged by their demonstrated curative effects on influenza and herpes simplex viruses [14], ZnO NPs have recently been investigated in vitro studies on human lung fibroblast cells [15]. The mechanism of the successful binding of ZnO NPs to COVID-19 targets through the formation of hydrogen bonds was described, highlighting the competence of ZnO NPs, which is strongly related to the effects of microstructure and morphology [15].

It is important to emphasize the key role of a synergistic experimental–theoretical approach to gain deeper insight into the mechanism of successful binding of ZnO. By integrating the synthesis of ZnO particles using different methods and experimental conditions and the measurement of structural and chemical properties, supported by theoretical calculations based on density functional theory (DFT), it is possible to prepare optimal ZnO samples with size- and morphology-dependent functionalities. Recent publications [16,17] have underlined the key role of an innovative aspect of a synergistic experimental–theoretical approach for a deeper insight into the growth mechanism and photocatalytic behavior of ZnO structures. The authors of [16] reported the photocatalytic efficiency of ZnO structures based on X-ray absorption spectroscopy (XAS) measurements, which showed the highest photocatalytic activity due to the presence of a larger amount of oxygen vacancies. In addition, a complementary understanding of the microstructure, formation mechanism and photocatalytic efficiency of differently prepared ZnO particles was obtained by integrating a complex XRD data analysis and DFT calculations. Due to the ability of ZnO NPs to generate reactive oxygen species by promoting H_2_O dissociation, which has been investigated by DFT calculations, new high-tech applications have emerged. The high affinity of H_2_O molecules for ZnO and the induced spontaneous dissociation of H_2_O were also reported by the Petridis research group for nearly spherical ZnO NPs with diameters ranging from 9 to 15 Å [18]. The surface arrangement of Zn and O atoms on the facets and edges of the nanoparticles, leading to adsorption sites with different local electronic structures, has been highlighted, providing non-equivalent sites for molecular or spontaneous dissociative adsorption. It is known that the surface of ZnO NPs is rich in –OH groups, which can be easily functionalized by various molecules [16,19,20].

The exactness of integrated experimental–theoretical approaches has recently increased considerably, enabling powerful computational tools for studying the behavior of metal oxides, such as ZnO [16,17,21]. The authors highlighted the importance of ethanolamines as stabilizers in the formation of ZnO and emphasized the usefulness of computational methods for understanding the electronic and optical properties of as-prepared ZnO. Among the various organic additives that have been used so far, a versatile family of ethanolamines (amino alcohols), which combine the properties of alcohols and amines, has found important applications because of their ability to bring about a significant change in particle surface properties [22,23,24,25]. Due to the ability of these types of organic additives to combine the properties of amines and alcohols and participate in reactions common to both groups, resulting in a significant change in the surface properties of ZnO particles, good control of their size and morphology can be achieved by changing the type of ethanolamines in the system according to the number of hydroxyl chains into mono- (EA), di- (DEA) and tri-ethanolamine (TEA). From our recent publications, as well as from a literature review [22,23,24,26], it appears that the aggregation of primary ZnO NPs into a sphere was achieved in the second stage of the formation process with a minimum of particle surface energy, confirming that the effect of the ethanolamine family on the aggregation process and the morphology of ZnO NPs depends on the number of hydroxyl branches. In addition, the relationship between particle size and optical properties of ZnO particles has been studied [17,26,27,28,29,30]. The effects of morphologically tuned nano- and micro-structured ZnO particles on photocatalytic activity are evaluated using calculations of the energy band gap from UV-Vis absorption spectra. The red shift in the absorption edge and the reduction in the band gap energies due to absorption closer to the visible region are clearly visible when the size of the ZnO particles increases.

Continuing our previous work describing the effect of the additive TEA [26,30], and to better explain the effects of adding the other two members of the amino alcohol family as additives, ZnO nanostructures are also prepared in media containing diethanolamine (DEA) and ethanolamine (EA). The aim of the present research is to explain the reactivity of ZnO nanostructures and the mechanism leading to their spherical aggregation based on the interaction between ZnO surfaces and additives through a joint experimental and DFT study. A thorough understanding of the mechanism of formation of spherical ZnO aggregates from ZnO nanoparticles and their stability by mimicking the processes at the computer level has yet to be achieved. The main objective of this systematic study is to improve the predictability of the aggregation behavior of ZnO nanoparticles into ZnO microspheres, especially their stability during aging times, with well-defined morphologies and physicochemical properties suitable for various applications, e.g., the adsorption and removal of various environmental pollutants in the presence of different amino alcohols, diethanolamine (DEA) and ethanolamine (EA).

## 2. Materials and Methods

### 2.1. Chemicals and Synthesis

Zinc acetylacetonate monohydrate, Zn(acac)_2_·H_2_O, (Zn(C_5_H_7_O_2_)_2_·H_2_O; 96%; Alfa Aeser^®^, Karlsruhe, Germany), diethanolamine (C_4_H_11_NO_2_; 98%; Sigma-Aldrich; Altötting, Germany), ethanolamine (C_2_H_7_NO; 99%; Sigma-Aldrich; Germany) and absolute ethanol (C_2_H_5_OH; 99.9% J. T. Baker; Deventer, the Netherlands) were used for the preparation of samples.

The experiments were divided into two groups. Considering our previously reported procedure [26], a similar synthesis strategy was used. The designation of the samples and the experimental conditions used for their preparation are listed in Table 1. The first group of samples was synthesized in the presence of diethanolamine (DEA), the second in the presence of ethanolamine (EA). In a typical synthesis procedure for this group of samples, diethanolamine (DEA) was first dissolved in 30 mL ethanol. Then, 0.5 g of Zn(acac)_2_·H_2_O was added to this solution, and the molar ratio of DEA to Zn(acac)_2_·H_2_O was adjusted to 1:1 (DEA/ethanol precursor solution of zinc salt). The transparent DEA/ethanol precursor solutions thus prepared were autoclaved at 170 °C for 4, 24 and 72 h (samples D4, D24 and D72, respectively). After autoclaving for an appropriate aging time in a Teflon-lined stainless steel autoclave, the obtained precipitates were centrifuged and washed several times with ethanol, then dried overnight under vacuum at room temperature. In the same way, the EA/ethanol precursor solutions of zinc salt were prepared and autoclaved at 170 °C for 4, 24 and 72 h (samples M4, M24 and M72, respectively).

### 2.2. Measurements and Characterization

A field-emission scanning electron microscope (FE-SEM), model JSM-7000F, manufactured by Jeol Ltd. (Akishima, Japan), was used for morphological analysis of the particles. UV/Vis absorption spectra were recorded using a Shimadzu (Kyoto, Japan) UV/Vis/NIR spectrometer with an integrated sphere (model UV-3600). The crystal structure of the samples was determined at room temperature using X-ray diffraction (XRD) on an Italstructures X-ray powder diffractometer (APD 2000, Cu-Kα radiation, graphite monochromator, scintillation detector).

### 2.3. Computational Methods

All geometry optimizations and calculations were performed by means of quantum chemical calculations at the density functional theory (DFT) level using the program Gaussian 09 (revision D1) package [31]. All optimizations and calculations were performed considering the ethanolic solvent effect (ethanol, ε = 24.85) using the implicit SMD (solvation model based on density) polarizable continuum solvation model [32]. For cluster modeling, a suitable (ZnO)_36_ cluster was used as a model for all possible molecular ZnO surface/DEA and ZnO surface/EA additive interaction predictions [33]. The (ZnO)_36_ cluster was constructed by Chen et al. [33], who also described the relationship between electronic properties and particle sizes. The M05-2X functional designed by Truhlar’s group was utilized [34]. For geometry optimization, the mixed 6-31+G(d,p) plus LANL2DZ basis set was used, which means, more precisely, that the 6-31+G(d,p) double-ξ basis set of Pople was chosen for the H, C, O and N atoms and the LANL2DZ basis was chosen for the transition-metal atoms (Zn) [35]. The final energies were refined using a 6-311++G(2df,2pd) basis set for the H, C, O and N atoms, while the same LANL2DZ-ECP basis set for zinc atoms was employed. The topological analysis of the charge density distribution was performed using Bader’s quantum theory of atoms in molecules (QTAIM) using the AIMALL (Version 17.01.25) software package [36,37]. The chemical bond nature can be described qualitatively regarding the signs and values of the electron density Laplacian, ∇^2^*ρ(r*_c_*)*, and of the electron energy density, *H(r*_c_*)*, at the corresponding bond critical point. The Gibbs free energies of interactions, ∆*G^*^*_INT_, were computed as the difference between the total free energy of the resulting structures (*G*^*^_AB_) and the sum of the total free energies (*G*^*^_A_ + *G*^*^_B_) of the associating units A and B, respectively. More computational details of the calculations can be found in Table 2.

## 3. Results and Discussion

### 3.1. Structure and Morphology

The microstructure and morphology of the ZnO particles were characterized using PXRD and FE-SEM. Regardless of the presence of different amino alcohols—diethanolamine (DEA) and ethanolamine (EA)—and the different aging times, the PXRD measurements show the formation of a hexagonal wurtzite ZnO structure with the space group symmetry *P*6_3_*mc*. Figure 1 shows the characteristic XRD patterns of the samples prepared in the presence of DEA or EA for different aging times between 4 and 72 h. As shown in Figure 1, the XRD pattern of sample D4 prepared in the presence of diethanolamine shows a significant broadening of the diffraction lines after 4 h of aging, indicating the presence of very fine ZnO particles. With an increase in aging up to 72 h, the broadening of the diffraction line decreased, indicating that samples D24 and D72 were relatively well crystallized. However, a significant broadening of the diffraction lines was observed in an M72 sample aged for 72 h in the presence of ethanolamine, indicating the presence of very fine ZnO particles.

The morphology of ZnO particles prepared in the presence of different amino alcohols, DEA and EA, for different aging times from 4 to 72 h was studied using FE-SEM (see Figure 2a–f) not only for their morphological characterization, but also for an explanation of their aggregation mechanism. The left panel shows ZnO particles prepared in the presence of ethanolamine, while the right side of Figure 2 shows SEM images of samples prepared in the presence of diethanolamine. The very well-defined, rounded nanoparticles, uniform in shape and nearly monodispersed, prepared in the presence of ethanolamine are shown in the left panel of Figure 2, with a mean particle size distribution of 8 nm (lower-left panel of Figure 2), while the right panel shows well-defined, rounded nanoparticles prepared in the presence of diethanolamine, with a mean particle size distribution of 15 nm (lower-right panel of Figure 2). The fine primary ZnO nanoparticles prepared in the presence of both types of amino alcohols, diethanolamine and ethanolamine, showed a tendency for spherical aggregation after four hours of aging, as shown in the SEM images in Figure 2a,d. Indeed, the large regular spheres with a size of ~500 nm dominate in both samples obtained after 4 h, regardless of the presence of different DEA or EA media. However, it can be seen that the ZnO nanoparticles were more densely aggregated in DEA media than in EA media. Moreover, the broken spheres and the much smaller, irregular, very weakly bound nanoparticle aggregates were also visible in the presence of EA (Figure 1a). These results suggest that the presence of DEA in an ethanolic solution could improve the DEA coating of ZnO nanoparticles, resulting in neighboring ZnO nanoparticles attracting each other more strongly, thereby leading to the formation of densely packed and more stable spherical aggregates compared to EA. However, the SEM image in Figure 2b shows that the primary ZnO nanoparticles (~10 nm), which assembled into large spherical aggregates after 4 h in EA, were easily separated after a longer aging time of 24 h and consisted of small irregular aggregates composed of a few loosely clustered nanoparticles. It is also clear that the small irregular ZnO nanoparticle aggregates obtained after 24 h were even more loosely interconnected and free-standing, and detached after an extended aging time of 72 h in EA media, as shown by the images from SEM in Figure 2c. At the same time, the spherical aggregates of ZnO nanoparticles prepared in the presence of DEA maintained their stability as dense spherical aggregates, even during prolonged aging of up to 72 h. The results seem to indicate that a longer aging time of 24 and 72 h could potentiate the DEA coating of ZnO nanoparticles and allow neighboring nanoparticles to cluster together more densely, leading to the formation of larger, densely packed and stable spherical aggregates, compared with EA (see Figure 2b,c,e,f).

### 3.2. UV-Vis Spectroscopy Analysis

It is important to note that the presence of the different amino alcohols, DEA and EA, at different aging times can change the size and morphology of the final ZnO particles, as well as their optical properties. There are some papers discussing the relationship between particle size and optical properties of ZnO particles [17,26,27,28,38]. It can be seen that the UV absorption edge is red-shifted, and the decrease in bandgap energies due to absorption closer to the visible region is clearly visible as the size of the ZnO particles increases [26]. The UV-Vis spectra of all prepared samples, the corresponding morphologies of the ZnO particles (insets) and the listed calculated bandgap energies obtained by the described method [39] are shown in Figure 3. The left panel shows the UV-Vis spectra of ZnO particles prepared in the presence of diethanolamine, while the right side of Figure 3 shows the UV-Vis spectra of samples prepared in the presence of ethanolamine. As shown in Figure 3, the ZnO samples had no optical absorption in the visible range, but there was good absorption in almost the entire UV region. It can be seen that the UV-Vis spectra of the ZnO samples prepared in the presence of DEA were characterized by intense absorption with two superimposed maxima, while the spectra of the samples prepared in the presence of EA were characterized by a maximum at around 350 nm, which had a slightly different shape and position. The effects of ZnO particles of different sizes and morphologies prepared in the presence of different amino alcohols on the optical properties were evaluated using energy band gap calculations and the UV-Vis absorption spectra. It can be seen that the calculated band gap energies for all samples were very close to each other. It appears that only small freestanding ZnO nanoparticles that detached after an extended aging time of 72 h in ethanolamine media were the reason for the blue-shifted absorption observed in the UV-Vis spectrum and some higher bandgap energies (E = 3.20 eV), as shown in Figure 3 (right panel, sample M72). J. Yu and X. Yu reported [40] some lower bandgap energies of hierarchical nanoporous ZnO hollow spheres, suggesting that ZnO with a smaller size has a larger redox potential for the photocatalytic degradation of organic pollutants under UV irradiation.

### 3.3. The Mechanism of Aggregation of ZnO Nanoparticles

Due to the different binding affinities that trigger the aggregation processes of ZnO nanoparticles into spherical aggregates in the presence of diethanolamine and ethanolamine, significantly different stability of the aggregates was achieved. The fine primary ZnO nanoparticles tended to achieve minimum particle surface energy by agglomerating into spherical aggregates immediately after their formation. Both amino alcohols, DEA and EA, might have contributed to slowing down the growth of the primary ZnO NPs in the first step and, at the same time, changing their surface energy by adsorption to the obtained primary NPs. Due to the potential of amino alcohols to combine the properties of alcohols and amines and simultaneously participate in reactions with both groups, leading to a significant change in the surface properties of ZnO nanoparticles, good control of spherical aggregation processes, particle size and morphology was achieved by adjusting the nature of amino alcohols in the system from mono-(EA) to di-(DEA) ethanolamine, depending on the number of hydroxyl groups. Indeed, the obtained primary ZnO nanoparticles showed a strong tendency for spherical aggregation after aging for 4 h in both media, DEA and EA. Moreover, it was found that the stability of spherical aggregates of ZnO nanoparticles in diethanolamine media was maintained, even up to 72 h of aging, which was not the case in EA media, as shown in the SEM images in Figure 2. To fully explain the aggregation mechanism and, in particular, the stability of the spherical aggregates of ZnO NPs, detailed theoretical studies of the interactions between the ZnO surface and additives using quantum chemical calculations are required to support the above experimental results. Based on microstructural and theoretical studies, the aggregation mechanism of ZnO NPs was proposed. The aggregation mechanism of spherical ZnO aggregates is inherently complex and suggests a profound effect of the surface interaction between neighboring ZnO NPs, which undoubtedly depends on the number of hydroxyl groups in DEA and EA. From our recent publications as well as the following literature review [22,23,24,25,26], the aggregation of primary ZnO NPs into a sphere was achieved in the second stage with a minimum of particle surface energy, confirming that the effect of the ethanolamine family on the morphology of ZnO NPs depends on the number of hydroxyl branches. The free hydroxyl branches have the ability to attract each other through hydrogen bonds and serve as a kind of “bridge” between neighboring ZnO NPs.

Understanding the mechanism of spherical ZnO aggregate formation and its stability by mimicking the processes at the computer level was achieved, in part, through theoretical simulations of the interactions between the ZnO surface and additives using a credible (ZnO)_36_–DEA and (ZnO)_36_–EA model, as shown in Figure 4. For cluster modeling, a suitable (ZnO)_36_ cluster was used, which was constructed by Chen et al. [33], who also reported the relationship between electronic properties and particle sizes. They found that the band gap of a spherical ZnO nanoparticle should be a linear function of the inverse of the particle diameter. DFT calculations indicate that the spherical aggregation of ZnO NPs is driven by the presence of various bonds such as coordinate bonds as well as noncovalent hydrogen bonds, van der Waals forces, electrostatic forces and so on. The calculated negative Gibbs free energy, ∆*G**_INT_, indicates that the (ZnO)_36_–DEA surface/additive interaction in diethanolamine media is a spontaneous exergonic process, with the calculated value ∆*G**_INT_ = −7.73 kcal mol^−1^. In ethanolamine media, however, the surface/additive (ZnO)_36_–EA interaction implies a structure for which the interactions represent an unfavorable, slightly endergonic process, with Δ*G**_INT_ = +0.50 kcal mol^−1^. The most stable (ZnO)_36_–DEA and (ZnO)_36_–EA structures are shown in Figure 4 and listed in Table 2 along with the bond lengths (*d*), energies (*E*) and QTAIM properties of the selected bonds of DEA and EA in the (ZnO)_36_ cluster for both structures. It is important to highlight the importance of the O–H∙∙∙O hydrogen bonds formed between the hydrogen atom of the hydroxyl groups of the DEA and the oxygen atoms in the (ZnO)_36_ cluster, as they are the only ones involved in the formation of a (ZnO)_36_–DEA structure in which both hydroxyl chains are bonded to the ZnO surface (Figure 4b). Due to the high affinity of the DEA molecules to the (ZnO)_36_ surface, and the high flexibility of the two hydroxyl groups on the other side, a high coverage level of the ZnO surface can be achieved. All remaining free hydroxyl groups of DEA as well as the free electron pair on the nitrogen atom have the ability to serve as a kind of “bridge” between neighboring ZnO NPs via O–H∙∙∙O- and N–H∙∙∙O-hydrogen bridges. The calculated values of the two O–H∙∙∙O hydrogen bonds are almost the same (*E*_O∙∙∙H_ ranges from −37.09 kcal mol^−1^ to −37.59 kcal mol^−1^, indicating short (d_O∙∙∙H_ = 1.401 Å) and unusually strong O–H∙∙∙O hydrogen bonding between the hydrogen atom of the hydroxyl groups of DEA and the oxygen atoms in the (ZnO)_36_ cluster. It seems that the presence of the two strong hydrogen bonds is the key factor for the very good and long-term aggregate stability of ZnO NPs in DEA media, which is consistent with the results of FE-SEM, Figure 2d–f.

When EA is used instead of DEA additive, the surface/additive interaction implies a (ZnO)_36_–EA structure for which the interactions represent an unfavorable, slightly endergonic process, with Δ*G**_INT_ = +0.50 kcal mol^−1^, revealing the special ability of the (ZnO)_36_ cluster to participate simultaneously in interactions involving both bonds, the coordinate (Zn–O) and the strong hydrogen bond (O–H∙∙∙O) (Figure 4b), which is more stable due to an additional weak N–H∙∙∙O hydrogen bond formed between the nitrogen atom of the amine group EA and the oxygen in the (ZnO)_36_ cluster (*E*_O∙∙∙H_ = −0.92 kcal mol^−1^). As a result of the (ZnO)_36_–EA interaction, the strongest binding was found to be via an extremely strong (E_O∙∙∙H_ = −51.81 kcal mol^−1^) and a particularly unusually short O–H∙∙∙O hydrogen bond (d_O∙∙∙H_ = 1.327 Å) between the hydrogen atom of the hydroxyl group of EA and the oxygen atom in the (ZnO)_36_ cluster, which was additionally accompanied by a new coordinate bond accomplished via a lone pair of electrons on the oxygen atom of the same hydroxyl group of EA with the zinc atom in the (ZnO)_36_ cluster (*E*_Zn―O_ = −19.36 kcal mol^−1^). The calculated energy values of *E*_O∙∙∙H_ (−51.81 kcal mol^−1^) and *E*_Zn―O_ (*E*_Zn―O_ = −19.36 kcal mol^−1^) indicate that the most stable (ZnO)_36_–EA structure possessed an incomparably stronger O–H∙∙∙O hydrogen-bonding ability, even more than twice that compared to the coordinate Zn–O bonding ability. Since the O–H∙∙∙O hydrogen bonding ability in the (ZnO)_36_–EA structure was more than twice as strong as the Zn–O coordinate bonding ability as well as incomparably stronger than the N–H∙∙∙O hydrogen bonding ability according to the calculation of DFT, it follows that the N–H∙∙∙O hydrogen bond, as well as the weaker Zn–O coordinate bond, was released more easily compared to the extremely strong O–H∙∙∙O bonds. Therefore, in the presence of EA, the primary ZnO NPs showed a strong tendency for spherical aggregation but no significant aggregate stability during the prolonged aging time, which is consistent with the results of FE SEM, Figure 2. Here, it is likely that the aging process first causes the decomposition of the large spherical ZnO aggregates, through a step of initial release of the weaker N–H∙∙∙O hydrogen bonds and perhaps the Zn–O coordinate bond than such a short and strong O–H∙∙∙O hydrogen bond. Moreover, it appears that due to the extremely strong O–H∙∙∙O hydrogen bonds, small irregular aggregates consisting of a few loosely clustered nanoparticles persist even when the weaker Zn–O coordinate and N–H∙∙∙O hydrogen bonds are released after an extended aging time of 24 h, which is consistent with the results of FE SEM, as shown in Figure 2a–c.

**Table 2 materials-16-04843-t002:** Bond lengths (*d*), energies (*E*), and QTAIM properties of the selected bonds in the most stable structures of the investigated systems in the ethanol solvent.

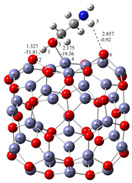 **EA–(ZnO)_36_**							
Bond	*d*/Å	*ρ(r_c_)*/ *e* × *a*_0_^−3^	∇^2^*ρ(r*_c_*)*/ *e* × *a*_0_^−5^	*V(r*_c_*)*/au	*G(r*_c_*)*/au	*H(r*_c_*)*/au ^a^	*E*/ kcal mol^−1 b^
H(1)-O(2)	1.327	1.208 × 10^−1^	−0.0065	−0.1651	0.0818	−0.0834	−51.81
O(3)-Zn(4)	2.175	4.621 × 10^−2^	0.1776	−0.0617	0.0531	−0.0087	−19.36
H(5)-O(6)	2.857	5.177 × 10^−3^	0.0186	−0.0029	0.0038	0.0009	−0.92
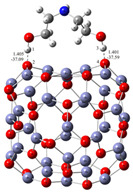 **DEA–(ZnO)_36_**							
Bond	*d*/Å	*ρ(r_c_)*/ *e* × *a*_0_^−3^	∇^2^*ρ(r*_c_*)*/ *e* × *a*_0_^−5^	*V(r*_c_*)*/au	*G(r*_c_*)*/au	*H(r*_c_*)*/au ^a^	*E*/ kcal mol^−1 b^
H(1)-O(2)	1.405	9.887 × 10^−2^	0.1052	−0.1182	0.0723	−0.0460	−37.09
H(3)-O(4)	1.401	9.965 × 10^−2^	0.1014	−0.1198	0.0726	−0.0472	−37.59

^a^ H(r_c_) = V(r_c_) + G(r_c_); ^b^ E = 0.5 × V(r_c_).

Notwithstanding the above-discussed parts of the formation and stability of spherical aggregates of ZnO nanoparticles, a more detailed DFT study of the interactions between the ZnO surface and the various amino alcohols is needed to fully elucidate the aggregation processes and, in particular, the stability of the aggregates. The strategy discussed above was to use the (ZnO)_36_–EA or (ZnO)_36_–DEA model, and the other was to add additional (ZnO)_36_ clusters mimicking the surface interactions between the neighboring ZnO nanoparticles in the presence of different amino alcohols by choosing an appropriate (ZnO)_36_–DEA–(ZnO)_36_ and (ZnO)_36_–EA–(ZnO)_36_ model. The most stable (ZnO)_36_–DEA–(ZnO)_36_ and (ZnO)_36_–EA–(ZnO)_36_ structures are shown in Figure 5.

The calculations revealed several interactions that could be useful in explaining aggregation processes, in particular, the better stability of ZnO aggregates in the presence of DEA compared to EA media. In particular, it is important to highlight the impact of the new, weak N–H∙∙∙O hydrogen bonding between the nitrogen atom of the amine groups in both amino alcohols (DEA and EA) and the oxygen in the additional (ZnO)_36_ cluster in (ZnO)_36_–DEA–(ZnO)_36_ and (ZnO)_36_–EA–(ZnO)_36_ structures on the formation and stability of the spherical ZnO aggregates. Thanks to the higher anchoring efficiency of diethanolamine to the ZnO surface due to two strong O–H∙∙∙O hydrogen bonds, higher and more stable surface coverage and stability were achieved compared to ethanolamine. At the same time, newly formed N–H∙∙∙O hydrogen bonds between free electron pairs at the nitrogen atom of the amine groups in DEA as well as EA and at the oxygen atom in the additional (ZnO)_36_ cluster have the ability to serve as a kind of “bridge” between the two (ZnO)_36_ clusters, as seen in the (ZnO)_36_–DEA–(ZnO)_36_ and (ZnO)_36_–EA–(ZnO)_36_ structures, respectively (Figure 5). The lower aggregation ability of neighboring ZnO nanoparticles and the lower stability of aggregates in ethanolamine media compared to diethanolamine media are probably a consequence of the higher energetic requirements during the aggregation process, which initially requires an additional step, to release the weak N–H∙∙∙O hydrogen bond in the (ZnO)_36_–EA structure, and immediately followed by the formation of a new N–H∙∙∙O hydrogen bond between free electron pairs at the same nitrogen atom of the amine groups in EA and at the oxygen atom in the additional (ZnO)_36_ cluster, bridging two (ZnO)_36_ clusters, as can be seen in the (ZnO)_36_–EA–(ZnO)_36_ structure (Figure 5a). Here, it is important to emphasize the possibility of the process of the release of the weak N–H∙∙∙O hydrogen bond in the unstable (ZnO)_36_–EA structure due to the access of the other additional (ZnO)_36_ cluster and the formation of new N–H∙∙∙O hydrogen bonds in the newly formed (ZnO)_36_–EA–(ZnO)_36_ structure. Considering all the above DFT results, it can be concluded that the surface interactions between the neighboring ZnO nanoparticles in the presence of different amino alcohols were spontaneously exergonic only in the presence of diethanolamine, while they were unfavorably endergonic in the presence of ethanolamine.

### 3.4. Comprehensive Discussion

In summary, the results of the DFT calculations of the interactions between the ZnO surfaces and additives using all (ZnO)_36_–EA and (ZnO)_36_–DEA (Figure 4) as well as (ZnO)_36_–DEA–(ZnO)_36_ and (ZnO)_36_–EA–(ZnO)_36_ (Figure 5) models are in excellent agreement with the microstructural results from FE SEM (Figure 2). A surface interaction between the neighboring ZnO NPs, depending on the number of hydroxyl chains of DEA or EA, allowed the nanoparticles to attach more tightly to each other in the presence of DEA than in the presence of EA, eventually leading to the formation of stable spherical aggregates even during prolonged aging of up to 72 h exclusively in the DEA medium (in agreement with the results of FE SEM, as shown in Figure 2). This was explained by the selection of an appropriate model for theoretical hypothetical simulations of the interactions between the ZnO surface and the additives, on the one hand, and between the neighboring ZnO nanoparticles, on the other hand, which is in agreement with the results of FE SEM. As described above, the first strategy is to use the (ZnO)_36_–EA or (ZnO)_36_–DEA model and the second is to add additional (ZnO)_36_ clusters that mimic the surface interactions between the neighboring ZnO nanoparticles in the presence of different amino alcohols by choosing an appropriate (ZnO)_36_–DEA–(ZnO)_36_ or (ZnO)_36_–EA–(ZnO)_36_ model. According to all DFT calculations, among the various existing parameters and interactions controlling the ZnO surface/additives interactions, the O–H∙∙∙O hydrogen bonds formed between the hydrogen atom of the hydroxyl groups of the amino alcohols and the oxygen atom in the (ZnO)_36_ cluster dominate and control the formation and stability of the spherical ZnO aggregates, regardless of the amino alcohols present, such as DEA or EA. Indeed, the presence of extremely strong hydrogen O–H∙∙∙O bonds in the (ZnO)_36_ cluster indicates a high but different chelating efficiency of various ethanolamines at surfaces, with DEA and EA apparently anchored by two or one hydrogen O–H∙∙∙O bonds. Since fine ZnO nanoparticles prepared in the presence of DEA or EA have different ratios of oxygen and hydroxide ions on this surface, the resulting ZnO aggregates formed by the different aggregation mechanisms described above definitely have different stability. Specifically, it is important to highlight the impact of the new, weak N–H∙∙∙O hydrogen bonds between the nitrogen atom of the amine groups in both amino alcohols (DEA and EA) and the oxygen in the additional (ZnO)_36_ cluster in (ZnO)_36_–DEA–(ZnO)_36_ and (ZnO)_36_–EA–(ZnO)_36_ structures as a „bridge“ between the neighboring ZnO nanoparticles, and the other stability-determining elements of the spherical ZnO aggregates. The main novelty is the correlation of the experimental results with computer calculations, which touches on one of the least studied research topics, namely, the determination of the mechanisms of degradation and the resistance of ZnO structures to experimental conditions, especially during aging times from 4 to 72 h. The results of the present study show a profound effect of surface interactions between the formed ZnO nanoparticles and ethanolamines on the way of aggregation of ZnO particles, which allows controlling their morphological properties. This study highlights the stability and compactness of spherical ZnO aggregates of ZnO NPs, prepared in the presence of diethanolamine compared to ethanolamine media, and provides a promising method and flexible design of ZnO nanomaterials to improve their adsorptive and optical properties.

## 4. Conclusions

This research highlights the great importance of an integrated experimental and theoretical approach to fully elucidate the mechanism of aggregation processes and, in particular, the stability and compactness of spherical aggregates of ZnO nanoparticles prepared in the presence of various amino alcohols—diethanolamine and ethanolamine—for a prolonged aging time. The results of the DFT calculations of the interactions between the ZnO surfaces and amino alcohols using all (ZnO)_36_/DEA and (ZnO)_36_/EA models are in excellent agreement with the microstructural results of FE SEM, which highlight the stability and compactness of spherical aggregates of ZnO nanoparticles prepared in the presence of diethanolamine. Due to the different binding affinities that trigger the aggregation processes of ZnO nanoparticles into spherical aggregates in the presence of diethanolamine and ethanolamine, significantly different aggregation mechanisms and stability of the aggregates were achieved. A surface interaction between the neighboring ZnO NPs, depending on the number of hydroxyl groups in DEA or EA, allowed the nanoparticles to attach more tightly to each other in the presence of DEA compared to EA, resulting in the formation of more stable and densely packed spherical aggregates, even during prolonged aging of up to 72 h in only the DEA medium, which is consistent with the results of FE SEM.

Thanks to the higher anchoring efficiency of diethanolamine to the ZnO surface due to two strong O–H∙∙∙O hydrogen bonds, higher and more stable surface coverage and stability were achieved compared to ethanolamine. The calculated negative free release energy ∆*G**_INT_ indicates that the interaction between the ZnO surface and amino alcohols in diethanolamine media was a spontaneous exergonic process (∆*G**_INT_ = −7.73 kcal mol^−1^), while in ethanolamine media, it is an unfavorable, slightly endergonic process (∆*G**_INT_ > 0). Since fine ZnO nanoparticles prepared in the presence of DEA or EA obviously have different ratios of oxygen and hydroxide ions on this surface, the resulting ZnO aggregates formed by the different aggregation mechanisms definitely have different stability. It is important to highlight the impact of the new, weak N–H∙∙∙O hydrogen bonds between the nitrogen atom of the amine groups in both amino alcohols (DEA and EA) and the oxygen in the additional (ZnO)_36_ cluster in (ZnO)_36_–DEA–(ZnO)_36_ and (ZnO)_36_–EA–(ZnO)_36_ structures as a „bridge“ between neighboring ZnO nanoparticles, and the stability-controlling elements of the spherical ZnO aggregates. It seems that the presence of two strong O–H∙∙∙O hydrogen bonds and, at the same time, a weaker N–H∙∙∙O hydrogen bond was the key factor for the very good and long-term aggregate stability of ZnO NPs in DEA media. This study highlights the stability and compactness of spherical ZnO aggregates of ZnO nanoparticles prepared in the presence of diethanolamine, and provides a promising method and flexible design of ZnO materials to improve its adsorptive and photocatalytic properties.

## Figures and Tables

**Figure 1 materials-16-04843-f001:**
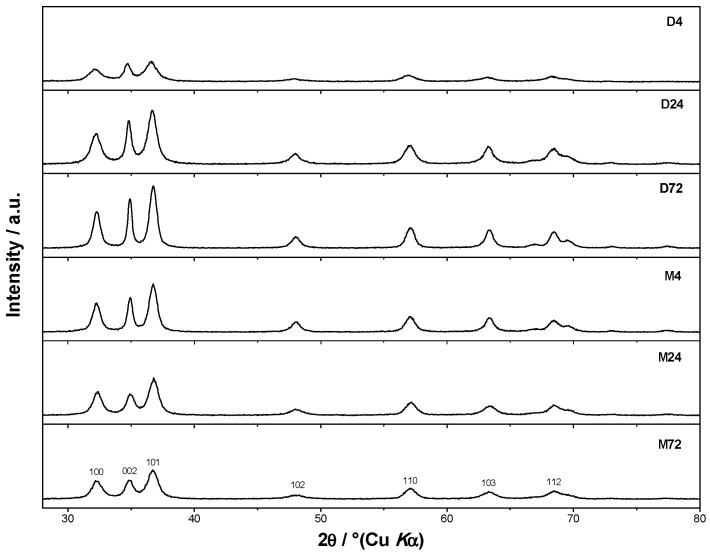
X-ray diffraction patterns of samples prepared in diethanolamine (**upper**) for 4 h (D4), 24 h (D24), 72 h (D72), and of ZnO samples prepared in ethanolamine (**below**) for 4 h (M4), 24 h (M24), 72 h (M72).

**Figure 2 materials-16-04843-f002:**
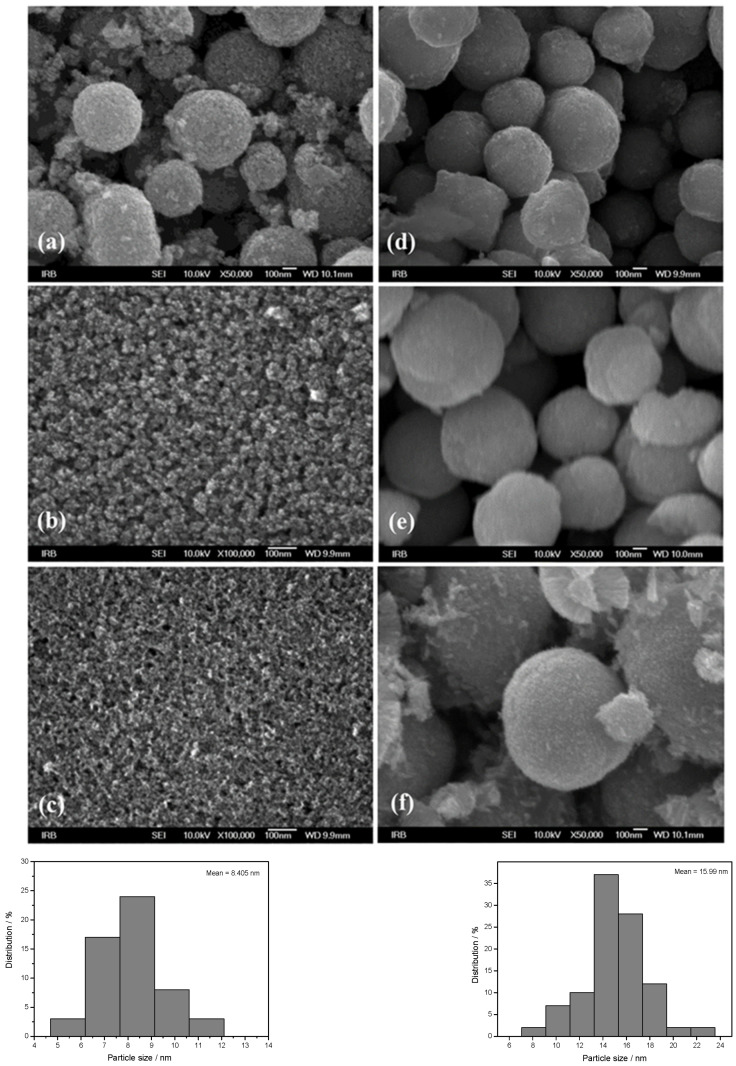
FE-SEM images of ZnO samples prepared in ethanolamine (**upper-left panel**): (**a**) M4, (**b**) M24, (**c**) M72, and of ZnO samples prepared in diethanolamine (**upper-right panel**): (**d**) D4, (**e**) D24, (**f**) D72, and particle size distribution: M72 (**lower-left panel**), D72 (**lower-right panel**).

**Figure 3 materials-16-04843-f003:**
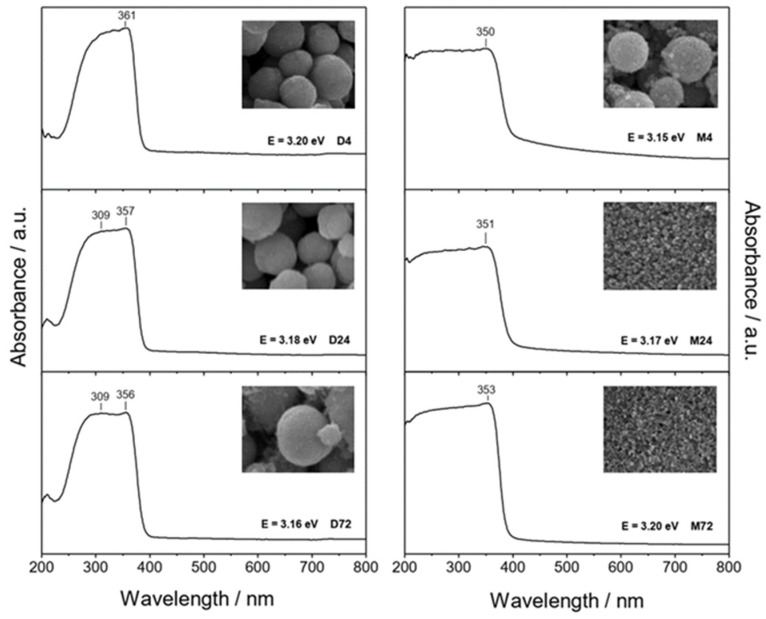
UV-Vis spectra of ZnO samples prepared in diethanolamine for 4 h (D4), 24 h (D24), 72 h (D72) (**left panel**), and in ethanolamine for 4 h (M4), 24 h (M24), 72 h (M72) (**right panel**). Inset images show the corresponding SEM images at different magnifications (D-samples: 50,000×, **left panel**, and M-samples: 100,000×, **right panel**) and the calculated band gap values.

**Figure 4 materials-16-04843-f004:**
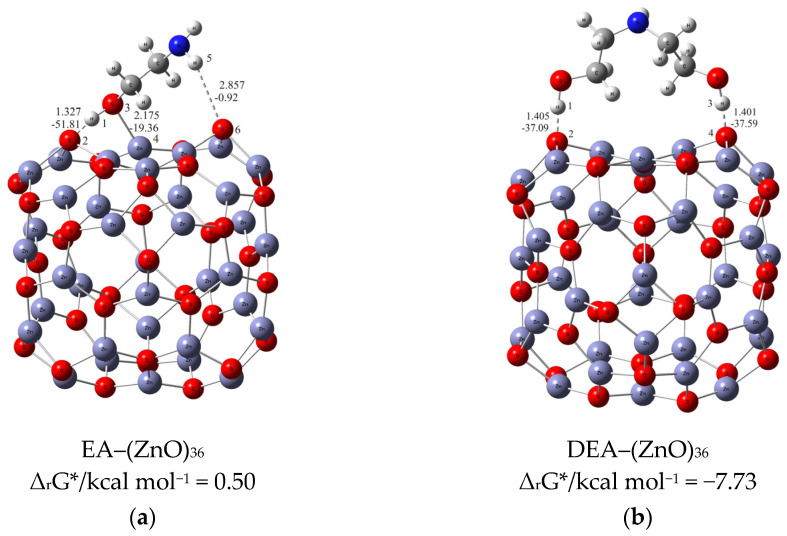
Most stable structures of (**a**) EA–ZnO)_36_; (**b**) DEA–(ZnO)_36_ (the bond distances are in Å, and the bond energies are in kcal mol^−1^).

**Figure 5 materials-16-04843-f005:**
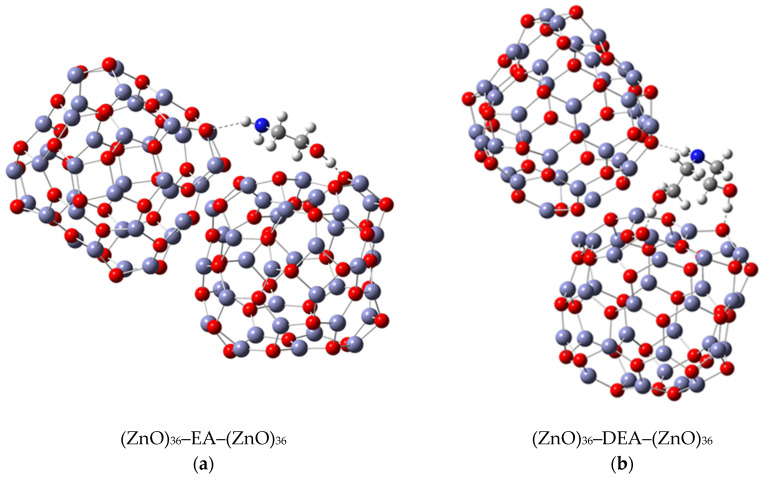
Most stable structures of (**a**) (ZnO)_36_–EA–(ZnO)_36_; (**b**) (ZnO)_36_–DEA–(ZnO)_36_DEA.

**Table 1 materials-16-04843-t001:** The notation and experimental conditions for the preparation of ZnO samples at an aging of temperature 170 °C.

Sample	(DEA)/(Zn(acac)_2_)	(EA)/(Zn(acac)_2_)	t_aging/h_
D4	1:1		4
D24	1:1		24
D72	1:1		72
M4		1:1	4
M24		1:1	24
M72		1:1	72

## Data Availability

The data presented in this manuscript will be made available from the corresponding author upon reasonable request.

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
