# Peer review of "Hydrogen Bonds as Stability-Controlling Elements of Spherical Aggregates of ZnO Nanoparticles: A Joint Experimental and Theoretical Approach"

_materials, 2023, doi:10.3390/ma16134843_

Round 1

Reviewer 1 Report

This is a very well-presented and exciting research. I congratulate the authors on their deep understanding and fluent presentation of complex chemical changes occurring during nanoparticle formation.

I have small things that came up upon reading:

1.      What is No in the following passage:

The authors highlighted the importance of ethanolamines as stabilizers in the formation of No and emphasized the usefulness of computational methods for understanding the electronic and optical properties of as prepared No.

2.      Please ensure you use uniform naming of the formed structures. For example, the spherical structures should not be called nanoparticles since they are much larger than 100nm, which seems to happen through the text.

Author Response

Journal:  Materials

Manuscript ID: materials-2477972

Title: “Hydrogen bonds as stability-controlling elements of spherical aggregates of ZnO nanoparticles: a joint experimental and theoretical approach”

Author(s): Ankica Šarić*, Ines Despotović

Response to Reviewer #1 Comments

This is a very well-presented and exciting research. I congratulate the authors on their deep understanding and fluent presentation of complex chemical changes occurring during nanoparticle formation.

Author's response:

Thank you very much for these comments and compliments.

I have small things that came up upon reading:

Point 1)

What is No in the following passage:

The authors highlighted the importance of ethanolamines as stabilizers in the formation of No and emphasized the usefulness of computational methods for understanding the electronic and optical properties of as prepared No.

Author's response: Thank you for noticing. There are two typo errors “No” which we have corrected to "ZnO" (on page 1, lines 77 and 79 in the revised manuscript)

Point 2)

Please ensure you use uniform naming of the formed structures. For example, the spherical structures should not be called nanoparticles since they are much larger than 100nm, which seems to happen through the text.

Author's response: For better clarity, we would like to discuss the used terminology of the formed structure. The fine primary ZnO nanoparticles (NPs) with a size of about 10 nm were prepared in the presence of both types of amino alcohols, diethanolamine and ethanolamine, which showed a tendency to spherical aggregation into the large regular spheres with a size of about 500 nm (SEM image in Fig. 2) For the large spherical aggregates of ZnO nanoparticles, larger than 100 nm, when the primary ZnO nanoparticles (~ 10 nm) were assembled, we used the term ”spherical aggregates of ZnO nanoparticles”.

Reviewer 2 Report

A very interesting article describing the effect of the solvents used (EA and DEA) on the stability and morphology of the spherical ZnO nanostructures.

Its greatest asset is the comparison of experimental results with computer calculations. Of course, this is the current trend that tries to explain the mechanism of synthesis of chemical structures (both in organic and inorganic chemistry). We must remember that such tools (in silico studies) are subject to imperfections and the synthesis itself may be sensitive to many factors that are difficult to take into account during calculations. However, in this case, such a comparison of both techniques is credible.

The experiment is properly planned and executed.

The authors used the appropriate techniques: synthetic, analytical and computational.

But as a practitioner using oxide supports to synthesize catalytic composites, I noticed the lack of BET measurements for obtaining ZnO structures in the body of the manuscript. The addition of data showing the results of BET measurements would be of interest to many scientists reading the text.

Line 33

“chemical stability” - ZnO is an amphoteric oxide and unfortunately its chemical stability (especially for low and high pH values) is poor. It should be specified what kind of chemical stability the Authors mean.

Line 75 and 76

“No” - incomprehensible. Did the authors mean "ZnO"?

Line 225

redundant dot after "increases" (before footnote [26])

Line 276-277 and line 446

“number of hydroxyl chains” - Are You sure it should be “chains”? Maybe “groups”?

Author Response

Journal:  Materials

Manuscript ID: materials-2477972

Title: “Hydrogen bonds as stability-controlling elements of spherical aggregates of ZnO nanoparticles: a joint experimental and theoretical approach”

Author(s): Ankica Šarić*, Ines Despotović

Response to Reviewer #2 Comments

A very interesting article describing the effect of the solvents used (EA and DEA) on the stability and morphology of the spherical ZnO nanostructures.

Its greatest asset is the comparison of experimental results with computer calculations. Of course, this is the current trend that tries to explain the mechanism of synthesis of chemical structures (both in organic and inorganic chemistry). We must remember that such tools (in silico studies) are subject to imperfections and the synthesis itself may be sensitive to many factors that are difficult to take into account during calculations. However, in this case, such a comparison of both techniques is credible.

The experiment is properly planned and executed.

The authors used the appropriate techniques: synthetic, analytical and computational.

But as a practitioner using oxide supports to synthesize catalytic composites, I noticed the lack of BET measurements for obtaining ZnO structures in the body of the manuscript. The addition of data showing the results of BET measurements would be of interest to many scientists reading the text.

Author's response:

Thank you for your kind comments and encouragement for further research. The authors agree with the Reviewer on the usefulness of BET measurements for ZnO nanostructures. This study highlights the poorly studied integrated experimental-theoretical approach to determine the mechanism of formation of spherical ZnO aggregates from ZnO nanoparticles and their stability by mimicking the processes at the computer level which has yet to be achieved. For further investigation, BET measurements will be performed.

Point 1)

Line 33

“chemical stability” - ZnO is an amphoteric oxide and unfortunately its chemical stability (especially for low and high pH values) is poor. It should be specified what kind of chemical stability the Authors mean.

Author reply: The authors agree with the Reviewer. The citation sentence related to stability, in particular “chemical stability” (on page 1, line 36 in the revised manuscript, yellow letters) has been rewritten: “Indeed, high thermal and mechanical stability and photostability are necessary conditions for the use of ZnO nanoparticles (NPs) as photocatalysts…”

Point 2)

Line 75 and 76

“No” - incomprehensible. Did the authors mean "ZnO"?

Author's response: Thank you for noticing. There are two typo errors “No” that we have corrected to "ZnO" (page 1, lines 77 and 79 in the revised manuscript)

Point 3)

Line 225

redundant dot after "increases" (before footnote [26])

Author's response: Redundant dot after "increases" (before footnote [26]) is deleted (page 7, line 233 in the revised manuscript)

Point 4)

Line 276-277 and line 446

“number of hydroxyl chains” - Are You sure it should be “chains”? Maybe “groups”?

Author's response: Thank you very much for your advice. Yes, “hydroxyl chains” is indeed a better than “number of hydroxyl chains” or “number of hydroxyl branches”. The part of sentences; “hydroxyl chains” and “hydroxyl branches” are replaced by “hydroxyl groups” on page numbers 8, lines 273 and 285 and page 13, line 461 in the revised manuscript.

Reviewer 3 Report

This article is devoted to studying the stability of nanoparticles based on zinc oxide, as well as determining the effect of various additives on the properties of nanoparticles. In general, this line of research is quite promising, as it touches on one of the poorly studied research topics related to determining the mechanisms of degradation and resistance of nanomaterials to external influences. At the same time, the authors of the work carried out a fairly large number of different studies, which allowed them to obtain a sufficient number of new unique results. In the opinion of the reviewer, this article fully corresponds to the subject of the journal and can be accepted for publication after the reviewers answer a number of questions that arose while reading this article.

1. When analyzing the morphology of the obtained spherical particles, the authors should pay attention to the difference in particle sizes, as well as give diagrams of the distribution and determination of the number of small grains and their nature.

2. The obtained X-ray diffraction data have a clear broadening, which can be due to both small particle sizes and their amorphous nature, and therefore the authors should pay attention to this phenomenon.

3. When analyzing the optical properties, the authors present only absorption spectra, but do not indicate such quantities as the band gap, refractive index, etc. Also, optical properties and their changes should be compared with data on structural parameters and dimensions, since the size effect can play a very important role in determining the optical characteristics of materials.

4. The authors should make a comparison of grain size data obtained by electron microscopy and X-ray diffraction.

5. The authors wrote in the abstract that the article will present the results of the change in the aging time on the mechanism of formation and stability of spherical aggregates of nanoparticles, however, these results are not presented explicitly.

Author Response

Journal:  Materials

Manuscript ID: materials-2477972

Title: “Hydrogen bonds as stability-controlling elements of spherical aggregates of ZnO nanoparticles: a joint experimental and theoretical approach”

Author(s): Ankica Šarić*, Ines Despotović

Response to Reviewer #3 Comments

 This article is devoted to studying the stability of nanoparticles based on zinc oxide, as well as determining the effect of various additives on the properties of nanoparticles. In general, this line of research is quite promising, as it touches on one of the poorly studied research topics related to determining the mechanisms of degradation and resistance of nanomaterials to external influences. At the same time, the authors of the work carried out a fairly large number of different studies, which allowed them to obtain a sufficient number of new unique results. In the opinion of the reviewer, this article fully corresponds to the subject of the journal and can be accepted for publication after the reviewers answer a number of questions that arose while reading this article.

Author's response:

Thank you for your kind comments and encouragement for further investigations.

The authors are grateful for your valuable and well-intentioned comments.

Point 1)

When analyzing the morphology of the obtained spherical particles, the authors should pay attention to the difference in particle sizes, as well as give diagrams of the distribution and determination of the number of small grains and their nature.

Author's response: Thank you for pointing this out. According to your suggestions given in point 1 we have made a change in Figure 2 and added new parts to Figure 2 (lower part) (page 6 lines 223 in the revised manuscript). Figure 2 was modified and has been supplemented by lower parts which show particle size distribution of ZnO samples prepared in EA and DEA media and new text has been added to the revised Manuscript to the caption for Figure 2 describing the new Figure 2:FE-SEM images of ZnO samples prepared in ethanolamine (upper left panel): (a) M4, (b) M24, (c) M72 and of ZnO samples prepared in diethanolamine (upper right panel): (d) D4, (e) D24, (f) D72 and particle size distribution: M72 (lower left panel), D72 (lower right panel)” (page 6 lines 224-226) To clarify the discussion of these results, a part of manuscript related to Figure 2 has been amended and new text has been added to the revised Manuscript in § 3. RESULTS AND DISCUSSION, Section 3.1. Structure and morphology (on page 5 line 192-197: in the revised manuscript, yellow font): „The very well defined, rounded nanoparticles, uniform in shape and nearly monodispersed, prepared in the presence of ethanolamine, are shown in the left panel of Figure 2, with a mean particle size distribution of 8 nm (lower left panel of Figure 2), while the right panel shows well defined, rounded nanoparticles prepared in the presence of diethanolamine, with a mean particle size distribution of 15 nm (lower right panel of Figure 2).“

Point 2)

The obtained X-ray diffraction data have a clear broadening, which can be due to both small particle sizes and their amorphous nature, and therefore the authors should pay attention to this phenomenon.

Author's response: Thank you for raising this important issue. We would like to discuss our results in the context of this comment. As shown in Figure 1, the XRD pattern of the sample prepared in the presence of diethanolamine shows a significant broadening of the diffraction lines after 4 hours of ageing, indicating the presence of very fine ZnO particles. With increasing ageing up to 72 hours, the broadening of the diffraction line decreased, indicating that samples are relatively well crystallized. However, a significant broadening of the diffraction lines was observed for sample aged for 72 hours in the presence of ethanolamine, indicating the presence of very fine ZnO particles. When the aging time is extended from 4 to 72 hours, a slight decrease in ZnO nanoparticles is observed, indicating their dissolution in the presence of ethanolamine. This observation indicates that a slight dissolution of ZnO particles occures in the presence of EA at pH values higher than the pH of the point of zero charge ZnO nanoparticles. To minimize the disssolution of ZnO particles all samples were prepared in the pH range of 9-11, depending on the added amount of EA.

Point 3)

When analyzing the optical properties, the authors present only absorption spectra, but do not indicate such quantities as the band gap, refractive index, etc. Also, optical properties and their changes should be compared with data on structural parameters and dimensions, since the size effect can play a very important role in determining the optical characteristics of materials.

Author's response: It is known that the size and morphology of ZnO particles have a significant effect on the optical properties of ZnO. For this reason, the effects of ZnO particles of different sizes and morphologies prepared in the presence of different amino alcohols on the optical properties were investigated by energy band gap calculations using UV-Vis absorption spectra (Figure 3, sample D4: E = 3.20 eV; sample D24: E = 3.18 eV; sample D72: E = 3.16 eV; sample M4: E = 3.15 eV; sample M24: E = 3.17 eV; sample M72: E = 3.20 eV). It can be seen that the calculated band gap energies for all samples are very close. It seems that only small free-standing ZnO nanoparticles which detached after an extended ageing time of 72 hours in ethanolamine media, are the reason for the blue shifted absorption observed in the UV-Vis spectrum and some higher bandgap energies (E = 3.20 eV) compared to samples M4 and M24.

Point 4)

The authors should make a comparison of grain size data obtained by electron microscopy and X-ray diffraction.

Author's response: It is important to emphasize that the magnification of 100.000x on these SEM pictures is very high.  Individual particles with a size of ~10 nm can be seen on the original images (which is close to the resolution limit of the instrument) which seems to correspond to the size of the crystallites.

Point 5)

The authors wrote in the abstract that the article will present the results of the change in the aging time on the mechanism of formation and stability of spherical aggregates of nanoparticles, however, these results are not presented explicitly.

Author's response: In order to clarify these results discussion, in particular the change in the aging time on the mechanism of formation and stability of spherical aggregates of nanoparticles, a part of manuscript regarding supposing mechanism of aggregation is supplemented and we added new text in the revised Manuscript at the end of section 3.4. Comprehensive discussion (on page 12 line 437-441 in the revised manuscript in yellow letters): “The main novelty is the correlation of experimental results with computer calculations, which touches on one of the poorly studied research topics related to the determination of the mechanisms of degradation and resistance of ZnO structures to experimental conditions, especially the ageing time from 4 to 72 hours.” We would like to discuss our results in connection with the comment, as was described in section 3.3. The mechanism of aggregation of ZnO nanoparticles, page 10, lines 347-356: “Therefore, in the presence of EA, the primary ZnO NPs showed a strong tendency to spherical aggregation but no significant aggregate stability during the prolonged ageing time, which is consistent with the results of FE SEM, Figure 2. Here, it is likely that the ageing process first causes the decomposition of the large spherical ZnO aggregates, through a step of initial release of the weaker N─H∙∙∙O hydrogen bonds and perhaps the Zn─O coordinate bond than such a short and strong O─H∙∙∙O hydrogen bond. Moreover, it appears that due to the extremely strong O─H∙∙∙O hydrogen bonds, small irregular aggregates consisting of a few loosely clustered nanoparticles persist even when the weaker Zn─O coordinate and N─H∙∙∙O hydrogen bonds are released after an extended ageing time of 24 hours, which is consistent with the results of FE SEM, Fig. 2 (a-c).”

Reviewer 4 Report

The manuscript is interesting and can be published after a minor revision as below.

1. The abstract does not reflects the study well, needs to be improved.

2. The quality of Figure 1 is low. 

3.  The novelty of the work should be highlighted.

4. The conclusion section is very long, the highlights of the study should be discussed in the conclusion section.

5. There are few typo errors needs to be rectified.

Author Response

Journal:  Materials

Manuscript ID: materials-2477972

Title: “Hydrogen bonds as stability-controlling elements of spherical aggregates of ZnO nanoparticles: a joint experimental and theoretical approach”

Author(s): Ankica Šarić*, Ines Despotović

Response to Reviewer #4 Comments

The manuscript is interesting and can be published after a minor revision as below.

Author reply:

Thank you very much for your kind and valuable comments.

Point 1)

The abstract does not reflects the study well, needs to be improved.

Author's response: Abstract section is improved. In the revised manuscript we have added the following new sentence in Abstract (page 1, line 24-26 in the revised manuscript): “The presence of two strong O─H∙∙∙O hydrogen bonds and at the same time a weaker N─H∙∙∙O hydrogen bond is the key factor for the very good and long-term aggregate stability of ZnO NPs in DEA media.” Just to mention, the abstract should be not exceed 200 words in total.

Point 2)

The quality of Figure 1 is low. 

Author's response: Thank you very much for pointing this out. In connection with checking the quality of Figure 1, we found that some of the resolution was lost when converting the image from the Origin to Word document to pdf file. For this reason, we have now submitted a new Figure 1, which has a better resolution.

Point 3)

The novelty of the work should be highlighted.

Author's response: Yes, that is indeed a good idea, that the novelty of the research could be additionally emphasized. In line with your suggestion in point 3, we have added new text in the revised Manuscript at the end of section 3.4. Comprehensive discussion on page 12, line 437-447 in the revised Manuscript, (in yellow font): “The main novelty is the correlation of experimental results with computer calculations, which touches on one of the poorly studied research topics related to the determination of the mechanisms of degradation and resistance of ZnO structures to experimental conditions, especially the ageing time from 4 to 72 hours. The results of the present study show a profound effect of surface interactions between the formed ZnO nanoparticles and ethanolamines on the way of aggregation of ZnO particles, which allows controlling their morphological properties. This study highlights the stability and compactness of spherical ZnO aggregates of ZnO NPs, prepared in the presence of diethanolamine compared to ethanolamine media, and provides a promising method and flexible design of ZnO nanomaterials to improve their adsorptive and optical properties.”

Point 4)

The conclusion section is very long, the highlights of the study should be discussed in the conclusion section.

Author's response: The 4. Conclusions section has now been shortened as much as possible. We are aware of the significant length of highlighting the major aspects of the paper in the Conclusion section, due of the DFT part related to the growth mechanism of ZnO nanostructures. Due to the reason that the computational chemistry part is of great benefit for the specific scientific community, we believe that the results described in the conclusions sections will be of broad interest, as they are of direct relevance to materials chemists and researchers working on ZnO nanomaterials for functional applications. Furthermore, we believe that presenting the conclusion in this way is necessary to provide a complete picture of the relevant results and to make them recognizable to a specific scientific community.

Point 5)

Comments on the Quality of English Language

There are few typo errors needs to be rectified.

Author's response: The typo errors and the English have been checked and corrected.

Round 2

Reviewer 3 Report

The authors answered all the questions posed, the article was accepted for publication.